# The Effect of Preventive Measures and Vaccination against SARS-CoV-2 on the Infection Risk, Treatment, and Hospitalization: A Cross-Sectional Study of Algeria

**DOI:** 10.3390/v14122771

**Published:** 2022-12-12

**Authors:** Ahmed Hamimes, Hani Amir Aouissi, Mostefa Ababsa, Mohamed Lounis, Umesh Jayarajah, Christian Napoli, Zaineb A. Kasemy

**Affiliations:** 1Faculty of Medicine, University of Constantine 3, Constantine 25000, Algeria; 2Scientific and Technical Research Center on Arid Regions (CRSTRA), Biskra 07000, Algeria; 3Laboratoire de Recherche et d’Étude en Aménagement et Urbanisme (LREAU), Université des Sciences et la Technologie (USTHB), Algiers 16000, Algeria; 4Environmental Research Center (CRE), Badji-Mokhtar Annaba University, Annaba 23000, Algeria; 5Department of Agro-Veterinary Science, Faculty of Natural and Life Sciences, University of Ziane Achour, Djelfa 17000, Algeria; 6Postgraduate Institute of Medicine, University of Colombo, Colombo 00700, Sri Lanka; 7Department of Medical Surgical Sciences and Translational Medicine, “Sapienza” University of Rome, Via di Grottarossa 1035/1039, 00189 Rome, Italy; 8Department of Public Health and Community Medicine, Faculty of Medicine, Menoufia University, Shibin El Kom 6131567, Egypt

**Keywords:** COVID-19, SARS-CoV-2, vaccination, preventive measures, Algeria

## Abstract

Coronavirus disease (COVID-19) caused by the SARS-CoV-2 virus continues to afflict many countries around the world. The resurgence of COVID-19 cases and deaths in many countries shows a complacency in adhering to preventive guidelines. Consequently, vaccination continues to be a crucial intervention to reduce the effects of this pandemic. This study investigated the impact of preventive measures and COVID-19 vaccination on the infection, medication, and hospitalization. A cross-sectional online survey was conducted between 23 December 2021 and 12 March 2022 in Algeria. To evaluate the effectiveness of strategies aimed at avoiding and minimizing SARS-CoV-2 infection and severity, a questionnaire was created and validated. Descriptive statistics and logistic regression analyses were computed to identify associations between dependent and independent variables. Variables with a *p*-value of < 0.05 were considered statistically significant. Our results indicated that out of 2294 answers received, only 16% of our sample was vaccinated, and more than 60% did not apply preventive guidelines. As a result, 45% were infected with SARS-CoV-2, 75% took treatment (even preventive), and 9% were hospitalized. The logistic regression showed that the impact of preventive measures on the unvaccinated is statistically not significant (OR: 0.764, 95% CI = 0. 555–1.052; *p* = 0.09). However, this relationship changes significantly for people who are vaccinated (OR: 0.108, 95% CI = 0.047–0.248; *p* < 0.0001). Our results also demonstrated that the impact of protective measures on non-vaccinated individuals is statistically significant in reducing the need to receive anti-COVID-19 treatments (OR: 0.447, 95% CI = 0.321–0.623; *p* < 0.0001). Furthermore, the results showed that the impact of preventive measures on the non-vaccinated population is also statistically significant in reducing the risk of hospitalization (OR: 0.211, 95% CI = 0.081–0.548; *p* < 0.0001). Moreover, vaccinated individuals who neglect preventive measures must take the COVID-19 medication at a rate of 3.77 times (OR: 3.77) higher than those who follow preventive measures and are vaccinated. In short, our findings demonstrate the importance of combining preventive measures and vaccination in order to fight against the pandemic. Therefore, we advise the Ministry of Health and relevant authorities to put more effort into enhancing public knowledge about the COVID-19 infection and vaccination through education and awareness initiatives. Parallel to implementing vaccination as additional preventive strategy, behavioral change initiatives must be improved to encourage adherence to COVID-19 prevention recommendations.

## 1. Introduction

With more than 624 million confirmed cases and 6.56 million deaths reported worldwide up to the second week of October 2022, it has been nearly three years since the world has started the fight against the coronavirus pandemic (COVID-19) [1].

In the Middle East and North Africa region (MENA) region, Arab countries reported more than 13 million cases with more than 172,000 deaths from COVID-19 [2].

The MENA region suffers from long-term structural challenges, including a low Gross Domestic Product (GDP) growth, high unemployment rate, low human capital index, small formal sector, poor foreign direct investment with a bad investment climate, low participation in global value chains, and rising debt levels [3]. All of the aforementioned pre-existing conditions, reflecting the fragile state of the social contract in many countries of the region, have increased the negative effects of COVID-19 [4].

Many prevention and control measures have been used to limit the spread of COVID-19, such as: shutting down travel and other activities; applying quarantine and screening measures, social distancing, and strict personal hygiene rules, such as frequent hand washing, disinfecting surfaces, and wearing face masks; avoiding close contact with potentially infected people; and staying at home when feeling sick, as suggested by previous scientific studies [5,6,7,8]. 

Moreover, there is currently a vaccination scheme to limit the spread of the virus and to minimize these costly measures against COVID-19, also reducing the indirect effect of the disease on both mental health- and health-related behaviors [9]. Nevertheless, until May 2022, almost one billion people in lower-income countries remained unvaccinated. Only fifty-seven countries have vaccinated at least 70% of their population; almost all of them are high-income countries. Vaccination in Arab countries ranges from 2 to 70%, and this wide range is due to war and political and financial instability [10]. There are other reasons behind the low vaccination rates, such as public concerns about the vaccine’s side effects, a poor understanding of COVID-19, and a mistrust in the country’s ability to combat it [11,12,13,14].

Moreover, while newly developed COVID-19 vaccines have shown high efficacy in clinical trials and excellent effectiveness in real-world data, some people still become infected even after vaccination [15]. Although the epidemiology of COVID-19 may change as new variants emerge, vaccination remains the safest strategy for averting future COVID-19 infections, hospitalizations, long-term sequelae, and death. Primary vaccination, additional doses, and booster doses are recommended for all eligible persons [16,17].

Regarding Algeria, the largest country in Africa, the fight against the virus has gone through numerous successive waves since the first confirmed case was declared on 25 February 2020 [18,19]. As of 10 October 2022, the country has recorded more than 27,100 confirmed cases and 6,881 deaths [20]. Not only are these numbers significant, they are also underestimated since the reported cases are based only on PCRs [21,22]. The national vaccination campaign started at the end of January 2021, with local policies targeting all vulnerable groups before being generalized to all populations [23]. However, the vaccination rate remains extraordinarily low, with only 17.9% of the population fully vaccinated. This is at odds with the country’s efforts to promote vaccination and represents a concern.

Since the epidemic’s onset, older age has frequently been associated with both a higher COVID-19 severity level and higher mortality risk [24,25,26]. Therefore, understanding the relationship between age and the compliance to preventive measures is necessary in order to better design the disease control campaign.

This study was conducted after the onset of the Omicron variant, when attitudes towards following the COVID-19 control measures and vaccination campaign declined [27], with the aim of analyzing the association between preventive measures and the COVID-19 vaccination, risk of infection, treatment, and hospitalization in Algeria.

## 2. Materials and Methods

### 2.1. Overview and Development of the Questionnaire

This study is an analytical cross-sectional study using a self-administered questionnaire (SAQ). To this end, this study was designed according to the STROBE guidelines for cross-sectional studies [28]. The survey was conducted from 23 December 2021 to 13 March 2022.

#### 2.1.1. Design

The questionnaire was designed with tools previously used to analyze attitudes and behaviors related to COVID-19 [7]. The questionnaire was reviewed by a panel of experts, including one epidemiologist, one public health professional, and one expert in infectious disease. Subsequently, the modified questionnaire underwent a preliminary test to ensure the intelligibility and the reliability of the items. The results confirmed that the content of the questionnaire was clear to the readers. The questionnaire was then stored in Google Forms (Google LLC, Menlo Park, CA, USA, 2021) and distributed online via email and social media platforms using Uniform Resource Locator (URL), and a quick response (QR) was sought. Participation in this survey was voluntary and without any incentive, and an e-consent was obtained from all participants before their enrollment.

The questionnaire was intended for all Algerian citizens over the age of 18 years who consented and completed the entire questionnaire. Given the size of the reference population, a sample of at least 385 participants was required to investigate the selected variables, assuming a response proportion of 50% and a 95% confidence level.

#### 2.1.2. Ethics

The study protocol was reviewed and approved by the Scientific Committee of the Scientific and Technical Research Center on Arid Regions (CRSTRA) (No 10/2021). The Declaration of Helsinki for research involving human subjects guided the conception and execution of the study [29]. All participants provided their informed consent digitally before completing the questionnaire.

### 2.2. Statistical Analyses

Initially, descriptive statistics such as frequencies (*n*) and percentages (%) were used to summarize the data using XLSTAT program, version 2020.1 (Addinsoft, Paris, France) [30]. A specific analysis on age was performed, given its relevance as a socio-demographic factor associated with the disease [24,25,26].

Next, to determine the association between preventive measures, vaccination, risk of infection, treatment, and risk of hospitalization, the variables were grouped, and a logistic regression method was used. All analytical tests were performed with a 95% confidence level (CI) and a significance level (Sig.) of ≤ 0.05. The JASP 0.14.1.0 program was used to analyze the data. JASP (0.14.1.0) is an open-source statistical analysis program supported by the University of Amsterdam [31]. It was designed to be easy to use and familiar to SPSS users. 

There are two types of variables: explanatory variables and response variables.

This questionnaire asked eight questions; three of which are basic questions and dependent variables (or response variables) used to measure the responses of individuals regarding the anticipated risks of COVID-19. The dependent variables are as follows:

$Y_{1}:$ Have you been infected with COVID-19? (Confirmed by PCR, serology, scan, antigenic test…).

$Y_{2}:$ Have you received any treatment? (Medicinal, natural, preventive, prescribed…).

$Y_{3}:$ Have you been hospitalized?

For the remaining five questions, we formulated two independent variables, X_1_ and X_2_, where each variable has only two values (0 or 1) for logistic regression, where X_1_ indicates non-vaccinated respondents and X_2_ indicates vaccinated respondents, as explained below.

With respect to the first independent variable:

$X_{1}=1:$ People applying preventive measures and who are not vaccinated. This category includes respondents who regularly wear a facemask, respect social distancing, follow strict hygiene protocols (hand washing/disinfection), and avoid gatherings (weddings, funerals, group prayers, religious ceremonies, birthdays…). In addition, this group is not vaccinated.$X_{1}=0:\;$ People not applying preventive measures and are not vaccinated. This category includes respondents who did not regularly wear a facemask, did not respect social distancing, did not follow strict hygiene protocols (hand washing/disinfection), or did not avoid gatherings (weddings, funerals, group prayers, religious ceremonies, birthdays…). In addition, this group is not vaccinated.

For the second independent variable:

$X_{2}=1:\;$ People applying preventive measures and who are vaccinated. This category includes respondents who regularly wear a facemask, respect social distancing, follow strict hygiene protocols (hand washing/disinfection), and avoid gatherings (weddings, funerals, group prayers, religious ceremonies, birthdays…) but who are vaccinated.$X_{2}=0:\;$ People not applying preventive measures and are vaccinated. This category includes respondents who did not regularly wear a facemask, did not respect social distancing, did not follow strict hygiene protocols (hand washing/disinfection), pr did not avoid gatherings (weddings, funerals, group prayers, religious ceremonies, birthdays…) but who are vaccinated.

In other words, there are two independent (explanatory) variables with binary responses which were recorded as follows:$$X_{1}=\left\{\begin{array}{ll} 1 & \mathrm{The}\;\mathrm{individual}\;i\;\mathrm{used}\;\mathrm{preventive}\;\mathrm{measures}\;\mathrm{and}\;\mathrm{was}\;\mathrm{not}\;\mathrm{vaccinated}\\ 0 & \mathrm{The}\;\mathrm{individual}\;i\;\mathrm{did}\;\mathrm{not}\;\mathrm{apply}\;\mathrm{preventive}\;\mathrm{measure}\;\mathrm{and}\;\mathrm{was}\;\mathrm{not}\;\mathrm{vaccinated} \end{array}\right.$$

And:$$X_{2}=\left\{\begin{array}{ll} 1 & \mathrm{The}\;\mathrm{individual}\;i\;\mathrm{used}\;\mathrm{preventive}\;\mathrm{measures}\;\mathrm{and}\;\mathrm{was}\;\mathrm{not}\;\mathrm{vaccinated}\\ 0 & \mathrm{The}\;\mathrm{individual}\;i\;\mathrm{did}\;\mathrm{not}\;\mathrm{apply}\;\mathrm{preventive}\;\mathrm{measure}\;\mathrm{and}\;\mathrm{was}\;\mathrm{not}\;\mathrm{vaccinated} \end{array}\right.$$

And:$$Z_{i}:\mathrm{age}\;\mathrm{of}\;\mathrm{individuals}\;\mathrm{associated}\;\mathrm{with}\;\mathrm{the}\;\mathrm{variable}\;X_{i}.$$

Thus, four groups were assigned:–People applying preventive measures and who are not vaccinated.–People not applying preventive measures and who are not vaccinated.–People applying preventive measures and who are vaccinated.–People not applying preventive measures but who are vaccinated.

## 3. Results

### 3.1. Descriptive Statistics

During the study period, a total of 3107 questionnaires were received, and incomplete responses were excluded. As a result, a sample of *n* = 2249 was selected for further analysis (Table 1).

The respondents were between the ages of 18 and 82 years old with a mean of 36.787 ± 14.177 (Table 2).

### 3.2. Results of the Logistic Regression

#### 3.2.1. The Relationship between Preventive Measures and Vaccination as Characterized by Probability of Infection

This survey studied the statistical association between the dependent variable, $Y_{1}$ (Have you been infected with COVID-19?), based on the variables $X_{1}$ (People applying preventive measures and who are not vaccinated versus people not applying preventive measures and who are not vaccinated) and $Z_{1}$ (age for both groups of the dichotomous variable). For comparison, we also studied the relationship between the same dependent variable $Y_{1}$ based on the variables $X_{2}$ (people applying preventive measures and who are vaccinated versus people not applying preventive measures but who are vaccinated) and $Z_{2}$ (age of individuals in this variable).

Both models were put into the following form:(1)PY1=1=11+exp−a1+β1X1++β2Z1(2)PY1=1=11+exp−a2+β1X2++β2Z2
where $a_{i}$ is a constant and $\beta_{i}$ is the parameter of the regression.

Based on Table 3 and Table 4, the impact of preventive measures on the non-vaccinated was not significant (Table 3). The odds ratio (OR: 0.76) suggests that individuals who were not vaccinated and who did not use preventive measures were at higher risk $\left(1/0.764=1.2\right)$ than those who were not vaccinated and applied preventive measures. However, this was not statistically significant. In Table 4, the role of preventive measures was important. The odds ratio (OR: 0.108) suggests that individuals who were vaccinated and did not respect preventive measures were at a higher risk $\left(1/0.108=9.26\right)$ than those who were vaccinated and applied preventive measures.

With lower BIC (Bayesian Information Criterion) scores, the summaries of the first two models show that $H_{1}$ suggests a significant relationship ($\chi^{2}$(841) = 26.216, p<0.0001) and (χ²(128) = 33.382, p<0.0001) successively for both models (Table 5 and Table 6).

Based on Figure 1, we can note the direct positive relationship between age and the dependent variable $Y_{1}$ in both models.

Figure 2 confirms the importance of preventive measures against COVID-19 infection for vaccinated people. This may be observed in the range between $X_{1}$ and $X_{2}$.

#### 3.2.2. Relationship between Preventive Measures and Vaccination as Characterized by COVID-19 Treatment

The association between the dependent variable $Y_{2}$ “Have you taken any COVID-19 treatment? (Medicinal, natural, preventive, prescribed…)” based on the variables $X_{1}$ (those who use preventive measures without vaccination versus those who do not use preventive measure without vaccination) and $Z_{1}$(age for both groups of the dichotomous variable) was studied. Furthermore, we also studied the relationship between the same dependent variable $Y_{2}$ based on the variables $X_{2}$ (people who apply preventive measures and are vaccinated versus people who do not apply preventive measures and are vaccinated) and $Z_{2}$ (age of the individuals in this variable).

Both models were put into the following form:(3)PY2=1=11+exp−a1+β1X1++β2Z1(4)PY2=1=11+exp−a2+β1X2++β2Z2
where $a_{i}$ is a constant and $\beta_{i}$ is the parameter of the regression.

Table 7 and Table 8 show that preventive measures are important for reducing the risk of having to take anti-COVID-19 treatments. Nevertheless, the impact of preventive measures was more concrete in a vaccinated environment.

With the lowest BIC scores, the summaries of models (3) and (4) show that $H_{1}$ suggests significant relationships of ($\chi^{2}$(841) = 58.059, p<0.0001) and ($\chi^{2}$(128) = 11.930, p<0.0001) successively for both models between the dependent variable ($Y_{2}:$“Have you taken COVID-19 treatment?”) and predictive variables (Table 9 and Table 10).

Based on Figure 3, the direct positive relationship between age and the dependent variable $Y_{1}$ in Model (3) was noted. In contrast, a flat line in model (4) indicated that age was not a significant variable in this model.

Figure 4 confirms the important role that preventive measures play in protecting an individual from having to take COVID-19 treatments in both vaccinated and unvaccinated environments.

#### 3.2.3. Relationship between Preventive Measures and Vaccination as Characterized by COVID-19 Hospitalization 

We studied the relationship between the dependent variable $Y_{3}$ “Have you been hospitalized ?” based on the variables $X_{1}$ (people who apply preventive measures without being vaccinated versus people who did not apply preventive measures without being vaccinated) and $Z_{1}$(age of the two groups of this dichotomous variable). Furthermore, we also studied the relationship between the same dependent variable $Y_{3}$ based on the variables $X_{2}$ (people who use preventive measures and are vaccinated versus people who do not use preventive measures but who are vaccinated) and $Z_{2}$ (age of individuals in this variable).

Both models are put into the following form:(5)PY3=1=11+exp−a1+β1X1++β2Z1(6)PY3=1=11+exp−a2+β1X2++β2Z2
where $a_{i}$ is a constant and $\beta_{i}$ is a regression parameter.

Table 11 indicates that individuals who do not apply preventive measures and are not vaccinated are at a risk of hospitalization that is $\left(1/0.211=4.74≅5\right)$ times higher than individuals who apply preventive measures without being vaccinated.

Table 12 indicates that the relationship between vaccinated individuals applying preventive measures versus vaccinated individuals who did not apply preventive measures is not significant concerning the risk of hospitalization. This supports the finding that the vaccination is a protective factor against hospitalization that limits the importance of protective measures as a preventive factor.

With the lowest BIC scores, the model summaries (5) and (6) show that $H_{1}$ suggests a significant relationship of (χ²(841) = 68.59; p<0.0001) and (χ²(128) = 11.26; p<0.0001) successively for both models between the dependent variable $Y_{3}$“Have you been hospitalized?” and the predictive variables (Table 13 and Table 14).

There is a strong relationship between age and the two explanatory variables. As age increases, the risk of hospitalization increases exponentially (Figure 5).

Figure 6 shows that preventive measures reduce the risk of hospitalization. In addition, vaccination is a crucial element in the fight against COVID-19. The figure on the right shows a low but parallel risk in the two groups of the second explanatory variable.

Table 15 show that the Area Under the Curve (AUC) parameter reached 0.5 in all models. Overall, a value of 0.5<AUC<1 suggests that there is a good probability that the classifier will be able to differentiate between positive- and negative-class values. Hence, the classifier can recognize more true positives and true negatives than false negatives and false positives.

The accuracy parameter measures the proportion of solutions that are relevant. This parameter measures the ability of the system to screen out irrelevant solutions. In our analysis, based on this index, the models used are relevant. Given the fact that the F-measure is greater than 0.5, we can conclude that our analysis is significant.

A Brier score was used for verifying the accuracy of a probability forecast. In our analysis, the Brier score was between 0.06 and 0.24. This indicates that the accuracy of the prediction is high.

## 4. Discussion

### 4.1. Significance of the Results

In the present work, an online survey-based study was carried out to evaluate the impact of preventive measures and COVID-19 vaccination on the possibility of infection, receiving anti-COVID-19 treatments, and hospitalization (due to COVID-19) in Algeria. Susceptibility to Severe Acute Respiratory Syndrome Coronavirus 2 (SARS-CoV-2) is universal, but older age has always been associated with disease severity. Nevertheless, our results showed that age was not significantly associated (*p* = 0.817 > 0.05) with the use of anti-COVID-19 drugs; this may be due to the heterogeneity of age among the sample. Preventive measures such as social distancing, quarantine, and facemask wearing have also been considered as key measures used to limit the spread of multiple diseases since a long time ago [32,33]. These measures were also adopted in the case of COVID-19 since it first appeared, especially before the first treatments and vaccines were introduced [34].

Our findings demonstrated that the role of preventive measures promoted by the World Health Organization (WHO) and the Centers for Disease Control and Prevention (CDC) was limited and insufficient for mitigating the spread of COVID-19 when people are unvaccinated. In fact, our study demonstrates that the influence of preventive measures on the unvaccinated is not statistically significant (OR: 0.764, 95% CI = 0. 555–1.052; *p* = 0.09). However, this relationship becomes significant for vaccinated people (OR: 0.108, 95% CI = 0.047–0.248; *p* < 0.0001). These results are consistent with the theory of herd immunity [35,36,37,38].

Furthermore, the study by Moore et al. [39] indicated that, while the spread of the virus in the United Kingdom (UK) was mitigated by introducing a set of social distancing measures, their influence was not sufficient to stop the spread of the epidemic. Given the continuing risk of another major infection, along with the negative socio-economic influence of preventive measures, vaccination was the proposed solution to mitigate the effects and spread of the global pandemic. According to Barajas-Nava [40], the ongoing transmission of the virus is mostly due to a lack of vaccines and insufficient infection control measures. Consequently, the role of vaccines remains central to the fight against the virus. Our results confirmed the need for raising awareness in the community about the expected consequences caused by the COVID-19 infection (even mild) in order to control the epidemic’s waves and to limit its spread by encouraging preventive measures and, overall, vaccination [11]. Similar results were also obtained by Masai and Akın [41] who followed reported cases of influenza-like illnesses with a sample of 758 symptomatic international students enrolled in universities in Turkey; they demonstrated that wearing facemasks, cleaning one’s hands, and adopting physical distancing can prevent the spread of a pandemic.

In this study, the results showed that the low efficacy of preventive measures for preventing infection in unvaccinated people does not necessarily mean that they are completely ineffective. As an example, our results show that protective measures on non-vaccinated individuals were statistically significant in reducing the need to receive anti-COVID-19 treatments (OR: 0.447, 95% CI = 0.321–0.623; *p* < 0.0001). These agree with the systematic review of Talic et al. [42], who found a reduction in the incidence of COVID-19 associated with mask-wearing, handwashing, and physical distancing. Thereby, preventive measures, even in the absence of vaccination, play a major role in preventing hospital admissions. Several studies have demonstrated the effectiveness of preventive measures alone, whether to fight against COVID-19, to reduce the spread of microorganisms in general, or to contribute to reducing incidences of other infectious diseases [43,44,45]. At the same time, COVID-19 is a major public health burden worldwide, and the morbidity and mortality worldwide are dramatically increasing [46]. Adherence to preventive measures against COVID-19 is still the best way to reduce and control the pandemic [47]. Generally, people with the best understanding and the best mental health are the most consistent in wearing a facemask, sanitizing their hands, practicing disinfection, and avoiding social groupings [48]. In our sample, only 23% respected social distancing, 35% followed strict hygiene protocols, and 46% wore a facemask; these low numbers may be explained by a lack of knowledge or the refusal to apply preventive guidelines.

In our study, non-vaccinated individuals who did not respect the preventive measures were more likely to take COVID-19 medication than those who followed the preventive measures (OR: 2.237). In general, herbal teas, aspirin, and dietary supplements based on zinc and vitamin C are prescribed in the case of a mild infection, even if they have no proven benefits [49,50]. This phenomenon may be represented in our study by the fact that 75% have taken supplements and/or natural remedies; however, only 16% of the sample was vaccinated, 45% was infected (almost exclusively unvaccinated), and, finally, 9% were hospitalized. Consequently, the role of preventive measures remains important also in the vaccinated community for the purpose of avoiding drug use. Despite many trials of drug repurposing, such as in the case of chloroquine (CQ) and hydroxychloroquine (HCQ) [51,52,53], some drugs, such as Veklury (Remdesivir) and Olumiant (baricitinib), were finally approved [54]. In addition, the development of oral antiviral drugs such as nirmatrelvir (Paxlovid^®^) and molnupiravir (Lavgevrio^®^) has shown promising clinical results for treating COVID-19 patients [55,56]. According to our results, vaccinated individuals who neglected the preventive measures took COVID-19 medication at a rate of 3.77 times higher than that of those who followed preventive measures and were vaccinated.

This study also demonstrated that protective measures are an important factor in preventing hospitalization. According to our results, the influence of preventive measures on the non-vaccinated is statistically significant in reducing the risk of hospitalization (OR: 0.211, 95% CI = 0.081–0.548; *p* < 0.0001). A recent study in Bangladesh confirmed the significant association (*p* = 0.0001) between COVID-19 preventive measures and the reduction of health outcomes, including infection, admission to hospitals, admissions to ICU wards, and death associated with COVID-19. On one hand, individuals who are not vaccinated and do not follow the preventative measures have a 4.74-fold higher risk of hospitalization from COVID-19 than those who follow the preventive measures and are not vaccinated. Vaccination, on the other hand, is the decisive factor for successfully avoiding hospitalization. Through this study, vaccination had a greater role in prevention of hospital admission than the preventive measures.

It is clear that the combination of vaccination and the preventive measures had a crucial role in preventing hospitalization [57]. In fact, according to Damjan et al. [58], in symptomatic patients with moderate vaccine coverage (between 40 and 70 %), vaccination was shown to be quite successful in preventing the disease from progressing to a more severe stage. Some studies reported the same finding. For example, a Turkish study that sampled 1401 adult patients in 25 hospitals showed that hospitalizations were lower in the group which received 2 doses of Sinovac and a booster dose of BioNTech compared to vaccinated patients with the same vaccines (as protection after two doses of Sinovac may not provide long-lasting immunity) and even moreso compared to hospitalizations of unvaccinated patients [59]. The same study showed that efficacy of vaccination may decrease over time; nonetheless, it may be enhanced by administering booster doses. In addition, it seems clear that effective vaccinations decreased COVID-19-related hospitalizations, and the importance of the booster dose is significant in this situation. The finding also suggests that vaccination can help to reduce the pressure on public health systems and, thus, benefits the overall public health community. Consequently, combining the benefits of both approaches will optimize protection and reduce the rate of hospitalizations.

Undoubtedly, further studies are required to comprehend the underlying mechanisms and the relationship between vaccination, preventive measures, and their specific effect on infection, public perceptions, and their influence on disease severity. In summary, this study constitutes a novel approach combining preventive guidelines and vaccination against SARS-CoV-2 infection and its severity.

### 4.2. Limitations

The present study has some limitations specifically related to the sample selection and the survey method. The survey was conducted using a convenient, snowball sampling method based on an online questionnaire that may have marginalized people without access to the internet and may also favor the overrepresentation of younger individuals who tend, generally, to spend more time on social media. While online recruitment guarantees large samples, it does not guarantee sample representativeness; thus, the generalizability of our findings to the national population may be limited.

### 4.3. Strengths

To our knowledge, this study is the first to provide evidence based on self-reported adherence to COVID-19 preventive measures and the vaccination’s influence on infection, treatment, and hospitalization among the Algerian population. The number of respondents (*n* = 2294) is quite high compared to almost all of the questionnaires presented in previous studies, specifically in the MENA region.

## 5. Conclusions

To the best of the authors’ knowledge, this is the first study that concerns COVID-19 preventive measures, vaccination, and their impact on infection, treatment, and hospitalization among the Algerian population. Our results showed that age was not significantly associated with the use of anti-COVID-19 measures; moreover, non-vaccinated individuals who do not respect the preventive measures are more likely to require COVID-19 medication than those who follow the preventive measures. Therefore, it is confirmed that the impact of preventive measures on the unvaccinated is statistically not significant; on the contrary, the correlation is significant for people who are vaccinated. At the same time, protective measures on non-vaccinated individuals are statistically significant in reducing both the need to receive anti-COVID-19 treatments and risk of hospitalization. 

In conclusion, this study confirmed that combining both vaccination and preventive measures is crucial in order to fight against the pandemic. Authorities must also develop effective, evidence-based health education programs and risk-communication campaigns that encourage citizens to change their habits, focusing particularly on those who do not follow recommended preventive measures, while encouraging vaccination at the same time.

## Figures and Tables

**Figure 1 viruses-14-02771-f001:**
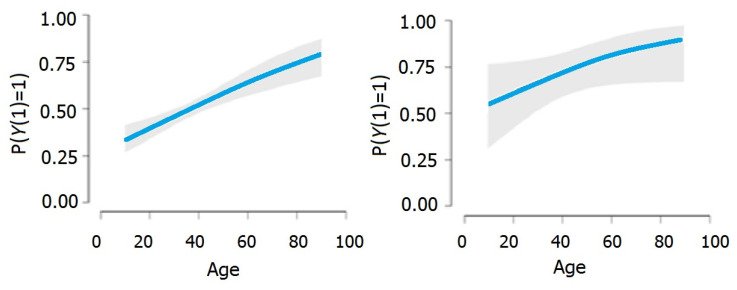
The relationship between age and the dependent variable $Y_{2}$ in both models.

**Figure 2 viruses-14-02771-f002:**
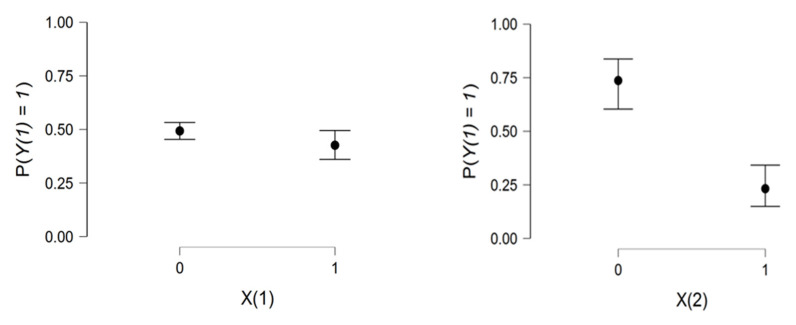
Relationship charts between the dependent variable $Y_{1}$ and the two explanatory variables $X_{1},X_{2}$.

**Figure 3 viruses-14-02771-f003:**
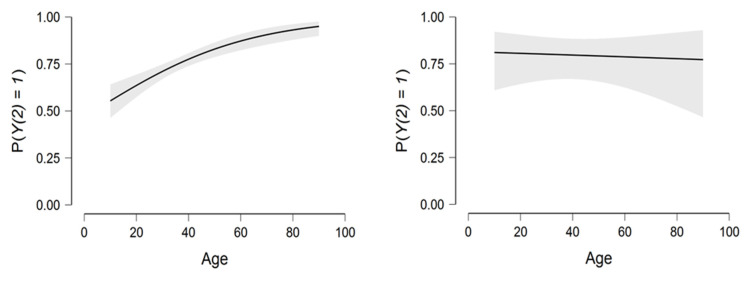
The relationship between age and the dependent variable $Y_{2}$ in both models.

**Figure 4 viruses-14-02771-f004:**
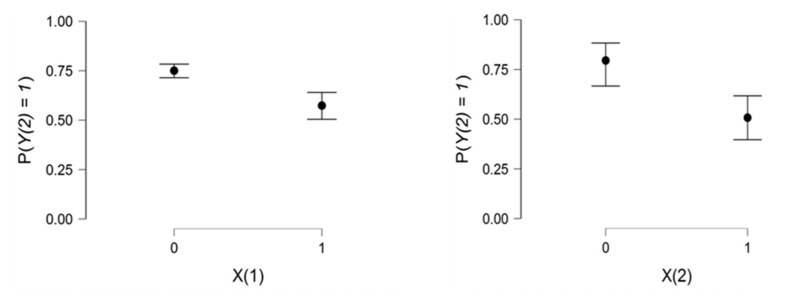
The relationship between age and the dependent variable $Y_{2}$ and the two explanatory variables $X_{1}\mathrm{and}X_{2}$.

**Figure 5 viruses-14-02771-f005:**
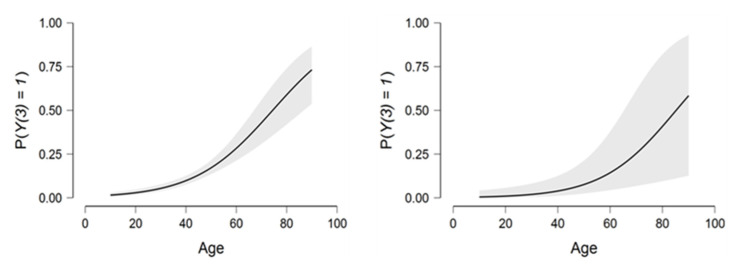
Relationship between the dependent variable $Y_{3}$ and the age variable.

**Figure 6 viruses-14-02771-f006:**
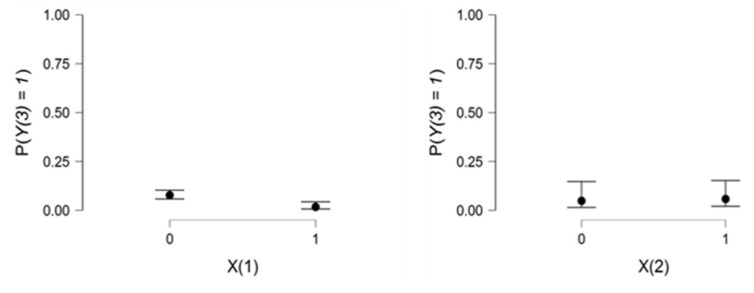
Relationship between the dependent variable $Y_{3}$
and the two explanatory variables $X_{1}\mathrm{and}X_{2}$.

**Table 1 viruses-14-02771-t001:** Summary of received responses.

Variable	Response	*n*	Percentage (%)
Did you wear a facemask regularly? (in public places, in the presence of other people)	No	1236	54%
	Yes	1058	46%
Did you respect social distancing? (2 m and above)	No	1755	77%
	Yes	539	23%
Did you follow strict hygiene? (Washing hands/disinfection)	No	1497	65%
	Yes	797	35%
Did you avoid clustering? (Weddings, funerals, group prayer, religious celebrations, birthdays…etc.)	No	1402	61%
	Yes	892	39%
Did you get vaccinated? (or at least 1 dose)	No	1933	84%
	Yes	361	16%
Have you been infected with COVID-19? (Confirmed by PCR, serology, scanner, antigenic…etc.)	No	1256	55%
	Yes	1038	45%
Have you taken any treatment? (medicinal, natural, preventive, prescribed…etc.)	No	575	25%
	Yes	1719	75%
Have you been hospitalized because of COVID-19?	No	2080	91%
	Yes	214	9%

**Table 2 viruses-14-02771-t002:** Age of the respondents.

Variable	Observations	Min	Max	Mean ± SD
Age ?	2294	18	82	36.787 ± 14.177

**Table 3 viruses-14-02771-t003:** The parameters of the logistic model characterizing the relationship between the variable $Y_{1}:$ “Have you been infected with COVID-19?” and the variables $\;X_{1}$ and $\;Z_{1}$ (Model 1). Bold values indicate a statistically significant impact.

Source	Value	SE	$\chi^{2}$	$\mathrm{Pr}\;>\;\chi^{2}$	OR	Lower CI (95%)	Upper CI (95%)
$a_{1}$	−0.931	0.219	18.090	<0.0001			
$\beta_{2}$	0.025	0.006	20.504	**<0.0001**	**1.025**	1.014	1.037
$\beta_{1}$	−0.269	0.163	2.718	0.099	0.764	0.555	1.052

**Table 4 viruses-14-02771-t004:** The parameters of the logistic model characterizing the relationship between the variable $Y_{1}:$ “Have you been infected with COVID-19?”and the variables $X_{2}$ and $Z_{2}$ (Model 2). Bold values indicate a statistically significant impact.

Source	Value	SE	$\chi^{2}$	$\mathrm{Pr}\;>\;\chi^{2}$	OR	Lower CI (95%)	Upper CI (95%)
$a_{2}$	−0.048	0.623	0.006	0.938			
$\beta_{2}$	0.025	0.014	3.291	0.070	1.025	0.998	1.053
$\beta_{1}$	−2.226	0.424	27.542	**<0.0001**	**0.108**	0.047	0.248

**Table 5 viruses-14-02771-t005:** Summary of the logistic model characterizing the relationship between the variable $Y_{1}:$ “Have you been infected with COVID-19?” and the variables $X_{1}$ and $Z_{1}$ (Model 1).

Model	Deviance	BIC	df	$\chi^{2}$	*p*-Value	McFadden R^2^	Nagelkerke R^2^
$H_{0}$	1168.136	1174.87	843				
$H_{1}$	1141.920	1162.13	841	26.216	<0.0001	0.022	0.041

**Table 6 viruses-14-02771-t006:** Summary of the logistic model characterizing the relationship between the variable $Y_{1}:$ “Have you been infected with COVID-19?” and the variables $X_{2}$ and $Z_{2}$ (Model 2).

Model	Deviance	BIC	df	$\chi^{2}$	*p*-Value	McFadden R^2^	Nagelkerke R^2^
$H_{0}$	179.883	184.758	130				
$H_{1}$	146.501	161.127	128	33.382	<0.0001	0.186	0.301

**Table 7 viruses-14-02771-t007:** The parameters of the logistic model characterizing the relationship between the variable $Y_{2}:$ “Have you taken any COVID-19 treatment?” and the variables $X_{1}$ and $Z_{1}$ (Model 3). Bold values indicate a statistically significant impact.

Source	Value	SE	$\chi^{2}$	$\mathrm{Pr}\;>\;\chi^{2}$	OR	Lower CI (95%)	Upper CI (95%)
$a_{1}$	−0.126	0.247	0.258	0.612			
$\beta_{2}$	0.034	0.007	25.948	**<0.0001**	**1.035**	1.021	1.048
$\beta_{1}$	−0.805	0.169	22.696	**<0.0001**	**0.447**	0.321	0.623

**Table 8 viruses-14-02771-t008:** The parameters of the logistic model characterizing the relationship between the variable $Y_{2}:$ “Have you taken any COVID-19 treatment?” and the variables $X_{2}$ and $Z_{2}$ (Model 4). Bold values indicate a statistically significant impact.

Source	Value	SE	$\chi^{2}$	$\mathrm{Pr}\;>\;\chi^{2}$	OR	Lower CI (95%)	Upper CI (95%)
$a_{2}$	1.482	0.615	5.808	0.016			
$\beta_{2}$	−0.003	0.012	0.053	0.817	0.997	0.973	1.022
$\beta_{1}$	−1.327	0.410	10.474	**0.001**	**0.265**	0.119	0.593

**Table 9 viruses-14-02771-t009:** Summary of the logistic model characterizing the relationship between the variable $Y_{2}:$ “Have you taken COVID-19 treatment?” and the variables $X_{1}$ and $Z_{1}$ (Model 3).

Model	Deviance	BIC	df	$\chi^{2}$	*p-*Value	McFadden R^2^	Nagelkerke R^2^
$H_{0}$	1035.840	1042.578	843				
$H_{1}$	977.780	997.995	841	58.059	<0.001	0.056	0.094

**Table 10 viruses-14-02771-t010:** Summary of the logistic model characterizing the relationship between the variable $Y_{2}:$ “Have you taken COVID-19 treatment?” and the variables $X_{2}$ and $Z_{2}$ (Model 4).

Model	Deviance	BIC	df	$\chi^{2}$	*p-*Value	McFadden R^2^	Nagelkerke R^2^
$H_{0}$	173.201	178.077	130				
$H_{1}$	161.272	175.897	128	11.930	0.003	0.069	0.119

**Table 11 viruses-14-02771-t011:** The parameters of the logistic model characterizing the relationship between the variable $Y_{3}:$ “Have you been hospitalized?” and the variables $X_{1}$ and $Z_{1}$ (Model 5). Bold values indicate a statistically significant impact.

Source	Value	SE	$\chi^{2}$	$\mathrm{Pr}\;>\;\chi^{2}$	OR	Lower CI (95%)	Upper CI (95%)
a	−4.797	0.436	121.232	<0.0001			
$\beta_{2}$	0.064	0.009	49.126	**<0.0001**	**1.067**	1.048	1.086
$\beta_{1}$	−1.556	0.487	10.204	**0.001**	**0.211**	0.081	0.548

**Table 12 viruses-14-02771-t012:** The parameters of the logistic model characterizing the relationship between the variable $Y_{3}$ “Have you been hospitalized?” and the variables $X_{1}$ and $Z_{1}$ (Model 6). Bold values indicate a statistically significant impact.

Source	Value	Standard Error	$\chi^{2}$	$\mathrm{Pr}\;>\;\chi^{2}$	OR	Lower CI (95%)	Upper CI (95%)
a	−6.036	1.334	20.484	<0.0001			
$\beta_{2}$	0.071	0.024	8.582	**0.003**	**1.073**	1.024	1.126
$\beta_{1}$	0.188	0.762	0.061	0.805	1.206	0.271	5.371

**Table 13 viruses-14-02771-t013:** Summary of the logistic model characterizing the relationship between the variable $Y_{3}$ “Have you been hospitalized?” and the variables $X_{1}$ and $Z_{1}$(Model 5).

Model	Deviance	BIC	Df	$\chi^{2}$	*p*-Value	McFadden R^2^	Nagelkerke R^2^
$H_{0}$	487.369	494.108	843				
$H_{1}$	418.778	438.993	841	68.591	<0.0001	0.141	0.178

**Table 14 viruses-14-02771-t014:** Summary of the logistic model characterizing the relationship between the variable $Y_{3}$ “Have you been hospitalized?”and the variables $X_{2}$ and $Z_{2}$(Model 6).

Model	Deviance	BIC	Df	$\chi^{2}$	*p*-Value	McFadden R^2^	Nagelkerke R^2^
$H_{0}$	75.550	80.425	130				
$H_{1}$	63.985	78.611	128	11.565	0.003	0.153	0.193

**Table 15 viruses-14-02771-t015:** The performance indicators of the used models.

Performance Metrics	Model (1)	Model (2)	Model (3)	Model (4)	Model (5)	Model (6)
AUC	0.583	0.774	0.664	0.658	0.786	0.783
Precision	0.586	0.722	0.719	0.657	-	-
F-measure	0.508	0.696	0.812	0.747	-	-
Brier score	0.242	0.186	0.197	0.214	0.071	0.068

## Data Availability

The data that support the findings of this study are available from the corresponding author upon reasonable request.

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
