# Peer review of "The Effect of Preventive Measures and Vaccination against SARS-CoV-2 on the Infection Risk, Treatment, and Hospitalization: A Cross-Sectional Study of Algeria"

_viruses, 2022, doi:10.3390/v14122771_

Round 1

Reviewer 1 Report

This was an interesting paper. Well done!

I have a few comments and questions that I would like you to address for each section:

Introduction:

It need a few information about what was highlighted in the aim or used in the analysis e.g. preventive measures, age…etc

More information about the context when the survey was distributed will make it better and clearer.

Methods:

This section need more information about validation of the questionnaire, the platform where it was hosted i.e. was it GDPR adherent, how it was distributed, and a brief description of its content. The way the analysis section was written included too many details that could be shortened, sample size calculation should be included, clarification why age only was analyzed out of other other socio-demographic factors.

Results:

Need to include a description of the sample as table in the results and  their applied preventive measures etc.

When describing results, it is better that the model used for analysis is not included only tables, figures and text description of them.

Discussion:

This is a cross sectional study-better not to say impact or effect.

Certain parts needed better linkage with results' explanation such as "Generally, people with the best knowledge and the best mental health are the most consistent...", " In general, herbal teas, aspirin and dietary supplements  based on zinc and vitamin C are prescribed in the case of a mild infection even if it has no proven benefits".

The explanation of the Turkish study need to be edited slightly to show the importance of the ref.

Why was a  a study about children included as your study focused primarily on adults. 

Conclusion:

No need to mention "The logistic regression" here

If you can shorten the conclusion to include the most important result and recommendation it might read better.

Good luck!

Author Response

We want to express our sincere gratitude to Reviewer #1 for the time dedicated to the review and the comprehensive, profound, and constructive remarks, which allowed us to improve the quality of our manuscript.

Please find attached a file with point-by-point responses to the reviewers' comments.

Reviewer 2 Report

Review of the paper titled: “The Effect of Preventive Measures and Vaccination against SARS-CoV-2 on the Infection Risk, Treatment and Hospitalization: A Cross-Sectional Study from Algeria”.

Abstract: line 37: “protective measures on non-vaccinated individuals is statistically significant in reducing the need to receive anti-COVID-19 treatments..” is or are? It’s a great mistake!

Introduction: line 57: what’s means GDP? Please, use for the first time the entire word: after, use the abbreviation. For the rest, this section is well written and clear.

Material and Methods: line 118: I think that it’s not necessary the ;. Please, correct.

Results: none

Discussion: the discussion is very well structured and written; highlights the strengths and weaknesses of the results of the manuscript and it is very interesting to note how vaccination acts as a watershed between the effectiveness or otherwise of the anti-SARS-CoV-2 prevention measures.

Author Response

We want to express our sincere gratitude to Reviewer #2 for the time dedicated to the review and the comprehensive, profound, and constructive remarks, which allowed us to improve the quality of our manuscript.

Please find attached a file with point-by-point responses to the reviewers' comments.
